# Recombinant Expression of Human IL-33 Protein and Its Effect on Skin Wound Healing in Diabetic Mice

**DOI:** 10.3390/bioengineering9120734

**Published:** 2022-11-28

**Authors:** Yunxian Li, Shixin Lin, Sheng Xiong, Qiuling Xie

**Affiliations:** 1College of Life Science and Technology, Jinan University, Guangzhou 510632, China; 2National Engineering Research Center of Genetic Medicine, Guangzhou 510632, China

**Keywords:** IL-33, diabetic wound healing, M2 macrophages, ILC2 cells

## Abstract

Chronic refractory wounds are one of the complications of diabetes mellitus that require effective therapy. The dermal-wound-healing property of IL-33 in diabetics is little understood. Therefore, this study aimed to express recombinant humanized mature IL-33 (rhmatIL-33) in *Escherichia coli* BL21 (DE3) and demonstrate its efficacy on dermal wounds in streptozotocin (STZ)-induced diabetic and nondiabetic mice by the dorsal incisional skin wound model. Results revealed that the rhmatIL-33 accelerated the scratch-healing of keratinocytes and fibroblasts at the cellular level. The wounds of diabetic mice (DM) showed more severe ulceration and inflammation than wild-type mice (WT), and the exogenous administration of rhmatIL-33 increased wound healing in both diabetic and wild-type mice. Compared with the up-regulation of endogenous IL-33 mRNA after injury in WT mice, the IL-33 mRNA decreased after injury in DM mice. Exogenous IL-33 administration increased the endogenous IL-33 mRNA in the DM group but decreased the IL-33 mRNA expression level of the WT group, indicating that IL-33 plays a balancing role in wound healing. IL-33 administration also elevated ILC2 cells in the wounds of diabetic and non-diabetic mice and improve the transcript levels of YM1, a marker of M2 macrophages. In conclusion, Hyperglycemia in diabetic mice inhibited the expression of IL-33 in the dermal wound. Exogenous addition of recombinant IL-33 promoted wound healing in diabetic mice by effectively increasing the level of IL-33 in wound tissue, increasing ILC2 cells, and accelerating the transformation of macrophage M1 to M2 phenotype.

## 1. Introduction

Wound healing is an orderly process that consists of three overlapping phases: inflammatory, proliferation, and remodeling [1]. Diabetic wound healing frequently often defies this schedule and stalls at certain stages due to permanent hyperglycemia [2]. Hyperglycemia, insulin resistance, and oxidative damage in diabetes can reduce the expression of various cytokines in wounds, such as the interleukin family, making wounds difficult to heal [3,4]. Studies have shown that many cytokines in the interleukin family are involved in wound healing, among which IL-33 plays an important role in inflammation, injury, and tissue homeostasis [5,6,7,8].

Diabetes is a multifaceted metabolic disease affecting more than 340 million individuals, and about 20% of them develop diabetic wounds [9]. In diabetic patients, leg or foot ulcers are the most common wounds that have a major impact on morbidity, disability, and mortality [10]. The underlying mechanisms of wound repair defects in diabetic patients are little understood, and the impaired expression of chemokines and other healing-related factors during the healing process is one of the reasons for delayed wound healing [11]. For non-healing wounds, the current standard of care includes wound debridement, avoidance of increased pressure on the injured site, repeated dressing changes, and antibiotics for infection [12]. Reports reveal that the standard treatments often lead to amputation, as more than 70% of diabetic wounds do not heal with current standard treatment strategies [13]. Few drugs are available to accelerate wound healing in clinical treatment [14]. Growth factors or cytokines provide continuous drug release in combination with dressings and exert local effects while limiting systemic side effects, promising clinical options for non-healing wounds [12].

Interleukin (IL)-33 is a newly discovered member of the IL-1 cytokine family [15]. The human full-length IL-33 protein includes 270 amino acids, called proIL-33, whose C-terminal is an IL-1-like cytokine domain. The N-terminal is a chromosome-binding domain, and the middle region contains many inflammatory protease cleavage sites [16]. After enzymatic cleavage, the IL-33 protein containing only the IL-1-like cytokine domain is produced, which is the highly active, mature form of IL-33 (matIL-33) that binds to the IL-33 receptor ST2 to exert its cytokine activity [15]. The pro-inflammatory role of IL-33 has been extensively studied in asthma, allergic reactions, and various inflammatory diseases [17,18,19]. However, it was recently found that IL-33 is also involved in the repair of tissue cell damage [20]. IL-33 is released from necrotic or damaged cells to function as an alarm protein [21]. Substantial recruitment of cells with strongly expressed IL-33 was found in skin wounds 24 h after injury in a rat model, and protein and mRNA levels of IL-33 were also elevated [22]. It was also found that the administration of exogenous IL-33 can promote wound tissue repair by promoting re-epithelialization and ILC2 proliferation [23]. Reports reveal that the recruitment and development of alternately activated macrophages have been observed at wound sites in IL-33-treated mice [24,25,26,27].

One of the reasons why the wounds of diabetic patients are difficult to heal is the impaired production of healing-related factors, such as growth factors and chemokines, during the healing process [3,4]. As an inflammatory factor, IL-33’s importance in wound healing in diabetic patients, and whether there are differences between the healing mechanisms of the normal wound and those of diabetic skin damage, remains unclear. In this study, a full-thickness skin defect model was designed in streptozotocin (STZ)-induced diabetic mice to assess wound healing. Comparing against wild-type mice, the therapeutic effect of IL-33 on wound healing in diabetic mice was evaluated by topical application of recombinant human mature IL-33(rhmatIL-33) to test the wound closure process.

## 2. Materials and Methods

### 2.1. Expression and Purification of rhmatIL-33 Protein in Escherichia coli

According to the nucleic acid sequence of human IL-33 in the Gene Bank database, the cDNA sequence (483 bp) of recombinant human mature IL-33 (rhmatIL-33) was obtained. The sequence was ligated into vector pET-28a (+) between the *NdeI* and *XhoI* sites to generate pET28a-rhmatIL-33, which contains a 6-histidine tag at the C-terminal for facilitating protein purification. The vector construction was performed by Suzhou Hongxun Biology Co. The recombinant plasmid pET28a-rhmatIL-33 was transformed into *E. coli* BL21 (DE3). The transformants were grown on Luria–Bertani (LB) agar plates containing 100 μg/mL kanamycin.

A single bacterial colony was picked from the transformant plate and inoculated into 5 mL of Luria–Bertani (LB) medium containing kanamycin (50 mg/mL) and subjected to agitation at 220 rpm, 37 °C for 12 h. A total of 2.5 mL of culture was transferred to 250 mL of fresh LB medium containing kanamycin (50 mg/mL) in a 1 L shake flask. Cultures were grown with 220 rpm shaking at 37 °C, then induced by adding 1 mM isopropylthio-β-D-galactoside (IPTG) at the OD600 of 0.6–0.8 for 6 h. Cells were harvested by centrifugation at 4000 rpm for 20 min and stored at −80 °C for further purification.

The cells (10 g) were resuspended in 100 mL of lysis buffer (20 mM imidazole, 1× PBS, pH 7.4), disrupted by sonication for 30 min on ice, and centrifuged at 12,000 rpm for 60 min. The supernatant was collected and filtered through a 0.45 μm filter. Ni–NTA column (Bestchrom Biotech, Shanghai, China) was used for affinity chromatography. Initially, the concentrated supernatant was poured through the column to allow protein binding. Next, the washing buffer (50 mM imidazole, 1× PBS, pH 7.4) was applied to wash away the contaminating proteins. Finally, the target protein was eluted by the elution buffer (1× PBS, pH 7.4) with different concentrations of imidazole (100 mM, 300 mM).

### 2.2. Cell Migration Assay

Both the human skin fibroblasts (HSF) cell line and the immortalized human keratinocyte (HaCaT) cell line were chosen for the cell migration test. Both cells were inoculated into 6-well plates at a density of 1 × 10^6^ cells/well and cultured in DMEM Medium (Gibco, Grand Island, USA) containing 10% fetal bovine serum (Excell Bio, Shanghai, China) at 37 °C in a humidified atmosphere containing 5%CO_2_ for 12 h. When the cell confluence reached 90%, the cells were starved for 2 h in a serum-free medium. Then, a 200 µL pipette tip was used to scratch the monolayer cells, and the samples were rinsed with PBS three times to remove non-adhered cells. Cells were then treated with various concentrations of rhmatIL-33 for 20 h, and DMEM with 1%FBS was used as the control. Images were taken at 0, 16, and 20 h after dosing. Cell migration distance was calculated by subtracting the scratch width at 16 and 24 h from the scratch width at 0 h.

### 2.3. Full-Thickness Excision Wound Model in Mice

C57BL/6 mice (20–25 g; 6–7 weeks old) were purchased from Guangzhou Qingle Life Science Co., Ltd. (Guangzhou, China) and singly housed in cages in a room at a constant temperature range (19~26 °C). All protocols were approved by the Laboratory Animal Ethics Committee of Jinan University (Ethics Code: 20210301-59).

The diabetic mice were induced as previously described [28]. Streptozotocin (STZ) normally makes mice diabetic by damaging their pancreatic islet cells. Diabetes was induced by intraperitoneal injection of 60 mg/kg streptozotocin (STZ) once daily for four consecutive days. Weight and blood glucose were measured five days after STZ injection to assess diabetes status in the animals. Routine weight and blood glucose assessments were performed weekly for three consecutive weeks. Mice with blood glucose levels ≥16.7 mmol/L were selected for the diabetic group.

The wild-type mice and the diabetic mice were randomly segregated into six groups (*n* = 9 mice per group): the uninjured group (WT + rhmatIL-33 and DM + rhmatIL-33), the solvent control group (WT and DM), and the rhmatIL-33 administration group (WT + rhmatIL-33 and DM + rhmatIL-33). After anesthesia with 3% pentobarbital sodium (45 mg/kg intraperitoneal), the dorsal hair of the mice was removed with a shaver and depilatory cream, followed by skin disinfection with 75% alcohol. A circular wound of 6 mm in diameter was made on the back with the lesion’s depth reaching the fascia layer by a 6 mm hole punch and ophthalmic scissors. A silicone ring with an inner diameter of 8 mm was used to limit the incised wound contraction.

### 2.4. Wound Assessment

The rhmatIL-33 (10 μL, 100 μg/mL) was administrated locally at the wound site once daily for seven days while using the same volume of PBS as the control. The day of the wound incision was recorded as day 0. From day 0 of injury, wounds were photographed every other day, the wound area was measured, and closure was recorded. The wound-contraction percentage was determined by the following equation: wound contraction (%) = (day 0 wound area − wound area on a particular day)/day 0 wound area × 100.

On days 5 and 7 post-injury, 3 mice were randomly killed in each group, and 2 mm wide skin tissue at the wound was excised and subsequently used for further analysis.

### 2.5. Real-Time PCR

Tissues were well ground with an electric grinder in RTL lysis buffer. Total RNA was extracted from wound tissues using HiPure Fibrous RNA Plus Kit (Magen, Guangzhou, China), and complementary DNA was then synthesized using PrimeScript 1st Strand cDNA Synthesis Kit (TaKaRa, Beijing, China) according to the manufacturer’s instructions. Next, qRT-PCR was performed with the ChamQ Universal SYBR^®^ qPCR Master Mix (Vazyme, Nanjing, China) on a Bio-Rad CFX384 qPCR cycler (Bio-Rad Laboratories, Shanghai, China) following the manufacturer’s instructions. With GAPDH as the internal reference gene, the relative quantification was carried out by the comparative CT method.

### 2.6. Flow Cytometry Assay

The tissue-digesting enzyme solution containing 100 μg/mL DNase I (Solarbio, Beijing, China) and 1 mg/mL collagenase IV (Worthington, Lakewood, LA, USA) was prepared in a cold serum-free DMEM medium and kept on ice for later use. The skin tissue was cut into pieces less than 1 mm by sterile ophthalmology scissors and digested with digestive enzyme solution (about 5 mL of enzyme solution for 1 cm^2^ of skin tissue) for 1.5–2 h in a steady-temperature incubator at 37 °C. Mouse skin tissue was filtered through a 70 µm cell strainer into PBS containing 2% FBS. The prepared cell suspension was directly used for staining and was fixed with 4% paraformaldehyde and stored at 4 °C.

Cell surface staining was performed with anti-CD45-BB515, anti-CD127-PE-CY7, anti-lineage-percp-cy5.5, and anti-ST2-BV421 (BD Pharmingen, California, USA), and incubated at 4 °C for 30 min in the dark. After washing with PBS, the cells were fixed with 4% paraformaldehyde and then permeabilized with 1× Perm/Wash Buffer (BD Pharmingen) for 15 min. Intracellular staining was performed with anti-GATA3-AF647 (BD Pharmingen). The cells were then incubated at 4 °C in the dark for 2 h and washed once with PBS. The data were acquired using a BD FACSCalibur flow cytometer (BD Bioscience, San Jose, CA) and analyzed with Flow Jo X.

### 2.7. Statistical Analysis

Statistical differences were determined using the one-way analysis of variance test of Student’s *t*-test. Data were expressed as mean and standard deviation. A value of *p* < 0.05 was considered statistically significant.

## 3. Results

### 3.1. Expression and Purification of rhmatIL-33 Protein

The rhmatIL-33 protein was recombinantly expressed in *E. coli* BL21(DE3). It was found that most of the rhmatIL-33 was retained in the supernatant in a soluble form after ultrasonication of cells. By Ni-affinity chromatography, the rhmatIL-33 was obtained in 100 mM and 300 mM imidazole eluates. The expression and purification of rhmatIL-33 were confirmed by Western blot analysis with anti-human IL-33 monoclonal antibody (Figure 1a). The purity of rhmatIL33 in 100 mM eluate was analyzed by SEC-HPLC and found to be 90–95%, which was used for subsequent experiments (Figure 1b).

### 3.2. Effects of rhmatIL-33 on the Migration of HSF and HaCaT Cells

Fibroblasts with contractile myofibroblast phenotype are essential for developing and remodeling connective tissue during skin-wound healing [29]. After skin injury, fibroblasts in the dermis began to proliferate, migrate to the wound, and activate at the wound to form an extracellular matrix that reshapes the wound bed [30]. The effect of rhmatIL-33 on the migration of human skin fibroblasts (HSF) was investigated by cell scratch assay in this study. The rhmatIL-33 was added at different concentrations, from 1 ng/mL to 40 ng/mL, while the PBS was used as the control group, and photographs were taken at three time points: 0 h, 16 h, and 20 h (Figure 2a). At 16 h, the scratch-healing rates of the groups with concentrations of 1 ng/mL, 5 ng/mL, 10 ng/mL, and 20 ng/mL were 87.69%, 85.37%, 94.83%, and 81.36%, respectively, which were significantly higher than that of the control group: 59.41% (*p* < 0.001). At 20 h, the scratch-healing rates of the groups with different concentrations of rhmatIL-33 were 94.22%, 94.80%, 99.00%, and 90.77%, respectively, which were also significantly higher than that of the control group: 66.76% (*p* < 0.001). However, a high concentration of rhmatIL-33 (40 ng/mL) could not improve cell migration (Figure 2b).

Keratinocytes are epidermal cells that produce keratin and play a crucial role in skin wound healing [31]. After skin injury, keratinocytes rapidly migrate to the wound area and proliferate to promote epithelial regeneration of the wound incision [32]. The effect of rhmatIL-33 on the migration of immortalized human keratinocytes (HaCaT) was also investigated. It was found that a higher concentration of rhmatIL-33 was needed to promote HaCaT cell migration; that is to say, rhmatIL-33 could promote the migration of HaCaT cells at concentrations of 20–160 ng/mL, while 1 ng/mL and 5 ng/mL rhmatIL-33 had no significant effects (data not shown). At 16 h, the scratch-healing rates of the groups with concentrations of 20 ng/mL, 40 ng/mL, 80 ng/mL, and 160 ng/mL were 59.66%, 59.79%, 59.36%, and 63.86%, respectively, and at 20 h, the scratch-healing rates were 65.78%, 74.79%, 68.49%, 69.74%, and 73.00%, respectively. The healing rates of all rhmatIL-33 treatment groups were significantly higher than that of the control group (53.43%) (Figure 2c,d).

### 3.3. Effect of rhmatIL-33 on Skin Wound Healing in Mice

A full-thickness skin defect model was used to evaluate the effect of rhmatIL-33 topical application on wound healing. After the full-thickness skin wounds on C57BL/6 mice, rhmatIL-33 (100 μg/mL, 10 μL) or PBS (negative control, 10 μL) was topically administered at the wound site once daily. In the wound model, wild-type mice were randomly divided into two groups (*n* = 9 mice/group): the negative control group (WT, wild-type) and the rhmatIL-33 treatment group (WT + rhmatIL-33). Likewise, diabetic mice were divided into the negative control group (DM, diabetes mellitus) and the rhmatIL-33 treatment group (DM + rhmatIL-33). Uninjured mice were used as a control group and divided into two groups: WT-Control and DM-Control.

In the WT group and WT + rhmatIL-33 group, no redness, swelling, or inflammation were found in the skin surrounding the wound. The wound-closure rate of the WT + rhmatIL-33 mice was significantly increased compared to that of the WT mice (*p* < 0.05). In the DM group, there was obvious ulceration and exudation on the third to fifth day of injury, and redness, swelling, and thick scabbing on the seventh day of injury; the DM + rhmatIL-33 group had no obvious redness, swelling, or inflammation. The wound of the DM + rhmatIL-33 group healed significantly faster than that of the DM group on the first, third, and ninth days of injury (*p* < 0.05) (Figure 3a,b). There was no effect on body weight in mice by topical administration of rhmatIL-33 (Figure 3c). These results indicated that compared with WT mice, the wounds of DM mice showed severe ulceration, inflammation, and other chronic wound characteristics, and administration of rhmatIL-33 can promote skin wound healing in wild-type and diabetic mice.

To further investigate the effect of rhmatIL-33 on in vivo wound healing, mice were sacrificed for histological analysis pre- and 5 days post-injury. The results of Hematoxylin-Eosin (HE) staining before injury showed that the skin epidermis in the DM-Control group was thinner than that of WT-Control group, indicating that the skin of the diabetic mice was more fragile than that of the wild-type mice (Figure 3e). The results of HE staining five days after injury showed that an intact skin stratum corneum with basal layer was seen in all groups except the DM group. Compared with the uninjured mice, in injured WT and DM mice, the injury resulted in enhanced inflammation (*p* < 0.0001). Compared with the non-administration group, the administration of rhmatIL-33 reduced the number of inflammatory cells in both the WT + rhmatIL-33 and DM + rhmatIL-33 groups (*p* < 0.0001) (Figure 3d–f).

### 3.4. Expression of Endogenous IL-33 in Skin Wounds

The qPCR results in wound tissue showed that the endogenous IL-33 mRNA in wild-type mice was gradually up-regulated over time after skin injury, indicating that the inflammatory response was triggered after injury. Contrastingly, the mRNA expression level of IL-33 in the WT + rhmatIL-33 group was down-regulated first and then up-regulated. Interestingly, the mRNA expression level of IL-33 in the DM group was also down-regulated first on the fifth day and then up-regulated on the seventh day, but with rhmatIL-33 administration, the IL-33 mRNA expression in DM + rhmatIL-33 group fluctuated according to the same pattern as with WT group. Overall, the IL-33 mRNA expression in diabetic mice was lower than that in wild-type mice, but the exogenous addition of recombinant rhmatIL-33 significantly increased the IL-33 mRNA expression in diabetic mice (Figure 4a).

An immunohistochemical study was performed to assess the expression of endogenous IL-33. (It has been verified that rhmatIL-33 protein does not cross-react with mouse IL-33 antibody.) The number of IL-33 positive cells was higher in the WT group than in the WT-control group (*p* < 0.001). Compared with the DM-control group, in the DM group, the number of IL-33 positive cells showed a downward trend, but there was no significant difference (Figure 4b,c). This indicated that the up-regulation of IL-33 was blocked in diabetic mice after injury compared with wild-type mice. After administration of rhmatIL-33, the number of IL-33 positive cells in the WT + rhmatIL-33 group was lower than that in the WT group (*p* < 0.05). However, the number of IL-33 positive cells in the DM + rhmatIL-33 group was higher than that in the DM group (*p* < 0.01) (Figure 4b,c).

### 3.5. Effects of rhmatIL-33 Administration on ILC2 Cells in Wound Tissue

Reports have shown that IL-33 promotes skin wound healing in mice by regulating ILC2 cells in tissues [33], and our study hypothesized that this mechanism was also applicable in diabetic mouse skin wounds. ILC2 cells were defined by CD45+, Lineage−, CD127+, ST2+, and GATA3+, and flow cytometry was performed to identify ILC2 cells in skin-wound tissues.

The results indicated that the percentage of ILC2s in skin wounds was significantly lower in DM mice than in WT mice no matter whether the mice were treated with rhmatIL-33. This partly explained why the skin of diabetic mice was difficult to heal after injury. On post-injury day seven, ILC2 cells were increased in all groups of mice, following IL-33 administration. In the wild-type mouse group, the percentage of ILC2s in skin wounds of the WT + rhmatIL-33 mice was significantly higher than that of WT mice (*p* < 0.001). In the diabetic mouse group, the percentage of ILC2 cells in the DM + rhmatIL-33 mice was also significantly higher than that of the DM mice (*p* < 0.05) (Figure 5a,b). These results indicate that ILC2 cells are involved in rhmatIL-33-promoting wound healing in both wild-type and diabetic mice.

The transition of macrophages from the pro-inflammatory MI type to the “alternatively activated” M2 type has been suggested to be necessary for the transition from inflammation to proliferation in the healing wound [34]. Activated ILC2 cells can produce IL-13 and IL-4, which are Th2-type cytokines that promote the transformation of macrophages to the M2-type, suppress inflammation, and participate in tissue repair [35,36]. Regardless of whether it was wild-type mice or diabetic mice, compared with the non-administered group, the expression of IL-13 in the exogenously rhmatIL-33-administered group showed an up-regulated trend, but there was no significant difference (Figure 5c); the expression of IL-4 in the WT + rhmatIL-33 group was significantly increased compared with the WT group (*p* < 0.05), but there was no significant difference between the DM + rhmatIL-33 group and the DM group (Figure 5d).

It has also been reported that ILC2s can promote macrophage polarization, leading to increased numbers of M2 macrophages [37]. Arg1 and YM1 are markers of M2-type macrophages. Compared with the untreated group, the mRNA expression of YM1 was significantly increased in wound tissues of the IL-33-treated group (Figure 5e); however, there was no significant difference in the mRNA expression level of Arg1 between them (Figure 5f). The results also indicated that the mRNA expression levels of Arg1 and YM1 in the wound tissue of diabetic mice were slightly lower than those of wild-type mice, while the magnitude of the increase in Arg1 and YM1 mRNA was greater in diabetic mice than in wild-type mice.

## 4. Discussion

During skin wound healing, early inflammatory responses occur, thereby enhancing repair and preventing infections by releasing cytokine signaling between immune and other skin cell populations [38]. Multiple findings demonstrate that IL-33 is a cytokine involved in repair events [23,24,25,26,27,33]. The pro-healing function of IL-33 was revealed in studies on keratinocyte epithelialization and alternate activation of macrophages [25,39]. However, the mechanisms underlying the action of IL-33 in diabetic wound healing remain unclear.

Results showed that the streptozotocin (STZ)-induced diabetic mouse had a weaker skin stratum corneum than wild-type mice, and diabetic mice exhibited severe wound ulceration and impaired wound healing. Exogenous recombinant human IL-33 accelerated wound healing in both wild-type and diabetic mice. The rhmatIL-33-treated mice exhibited a lower number of inflammatory cells.

In wild-type mice, injury led to up-regulation of endogenous IL-33 expression in mouse skin wounds, but the up-regulation of IL-33 expression was blocked after skin injury in diabetic mice. The expression of endogenous IL-33 in the skin wound tissue of diabetic mice was significantly up-regulated after rhmatIL-33 administration. However, endogenous IL-33 decreased when exogenous IL-33 was added to wild-type mice, indicating that IL-33 plays a balancing role in wound healing. This study also demonstrated that the proportion of ILC2s in the skin wounds of the DM mice was significantly lower than that of the WT mice, and IL-33 significantly increased the proportion of ILC2s in skin wounds in wild-type and diabetic mice.

Studies have shown that the expression of IL-33 is significantly increased in the skin tissue of infectious lesions in mice, which may be related to the release of a large amount of IL-33 after local tissue injury [2,40]. Our study indicated that the transcription and expression of endogenous IL-33 in wild-type mice increased after injury, but the endogenous IL-33 in WT mice showed a downward trend after the addition of exogenous IL-33. In contrast, the level of endogenous IL-33 in DM mice did not increase but instead decreased after injury, and after the administration of recombinant IL-33, the endogenous IL-33 in the wound tissue of the diabetic mice gradually increased with the same pattern as in the WT mice. High blood glucose is believed to inhibit the expression of IL-33 [38]; thus, the expression and secretion of IL-33 could not be induced due to skin injury in DM mice as it was in wild-type mice, which may be one of the reasons why the wounds of DM mice are difficult to heal. It has been reported that hyperglycemia suppresses the expression of IL-33 in a diabetic mouse wound excision model and decreases the expression of REG3A in keratinocytes, thereby increasing inflammation in skin wounds [41]. It was also found that hyperglycemia inhibits the expression of IL-33 in mouse myocardium, thereby aggravating ischemia/reperfusion-induced myocardial injury in diabetic mice. Additionally, exogenous IL-33 treatment attenuated hypoxia-induced injury in diabetic mice [42]. However, it has been reported that IL-33 expression is increased in primary hepatocytes treated with high glucose [43].

It has been reported that IL-33 can promote wound healing in an acute setting by promoting the proliferation of ILC2s and re-epithelialization [33]. ILC2s are present in healthy human and murine skin, and their numbers are elevated in the presence of skin inflammation [44]. Our study indicated that the proportion of ILC2s in WT mice was significantly higher than in DM mice, and ILC2s in the wound tissue of both WT and DM mice were significantly increased after topical rhmatIL-33 administration. These results confirm that IL-33-induced ILC2 cells play an important role in wound healing.

The skin damage repair process includes two stages. In the early stage, M1 macrophages mediate tissue damage and trigger an inflammatory response [45]; in the later stage of wound healing, infiltrating macrophages express the M2 phenotype, limit the inflammatory response, promote angiogenesis, and participate in the remodeling and repair of damaged tissue [46,47]. IL-33, an IL-1-like cytokine that signals through the IL-1 receptor-related protein ST2, induces T-helper type 2-associated cytokines (IL-4, IL-5, and IL-13, etc.) and participates in various inflammatory and immune responses [35,36]. IL-33 cooperates with IL-13 to induce M2 polarization of macrophages and promote wound repair. Markers such as YM1 and Arg1 are up-regulated during M2 polarization [23,28]. This study indicates that exogenous IL-33 contributes to the expression of IL-13 and IL-4 in wound tissue, and the expression of M2 macrophage marker YM1 is significantly increased, indicating that IL-33 plays an important role in the transition of macrophages from M1 to M2 in wound healing. Contrastingly, Arg1, another M2 macrophage marker, was not significantly up-regulated by the addition of exogenous IL-33. A recent study found that an increase in YM1 expression may act to control or limit the activation of the M2 macrophage marker Arg1 during macrophage polarization [48].

As illustrated schematically in Figure 6, IL-33 is a coordinating factor in the wound-healing process and can promote wound healing by enhancing ILC2 cells and accelerating the transformation of macrophages from the M1 to M2 phenotype. Hyperglycemia in the damaged skin tissue of diabetic mice can inhibit the expression of IL-33, which is one of the reasons why skin wounds in diabetic patients are difficult to heal. Exogenous recombinant IL-33 effectively increased the level of IL-33 in wound tissue, thereby promoting wound healing in diabetic mice.

## Figures and Tables

**Figure 1 bioengineering-09-00734-f001:**
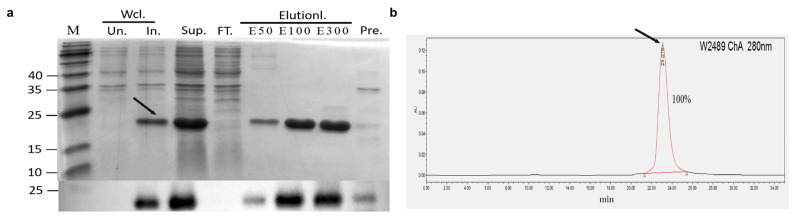
Expression and purification of rhmatIL-33 protein. (**a**) Expression and purification of the rhmatIL-33 protein were checked on western blotting and 15% SDS-polyacrylamide gel electrophoresis (SDS–PAGE). Aliquots from cell cultures of strain BL21(DE3) grown at 37 °C were taken before and after induction (un and In). Cells were harvested 6 h after induction and disrupted by sonication for 30 min on ice; the expression of rhmatIL-33 in the supernatants and pellets was detected (Sup and Pel). The supernatants were applied to Ni–NTA affinity chromatography and eluted with 50 mM, 100 mM, and 300 mM imidazole, respectively (Lanes E50, E100, E300). Lane FT., Flowthrough; Lane M, molecular weight standards (kDa) (**b**) Analysis of purified rhmatIL-33 by SEC-HPLC. The arrow indicates the position where rhmatIL-33 is eluted.

**Figure 2 bioengineering-09-00734-f002:**
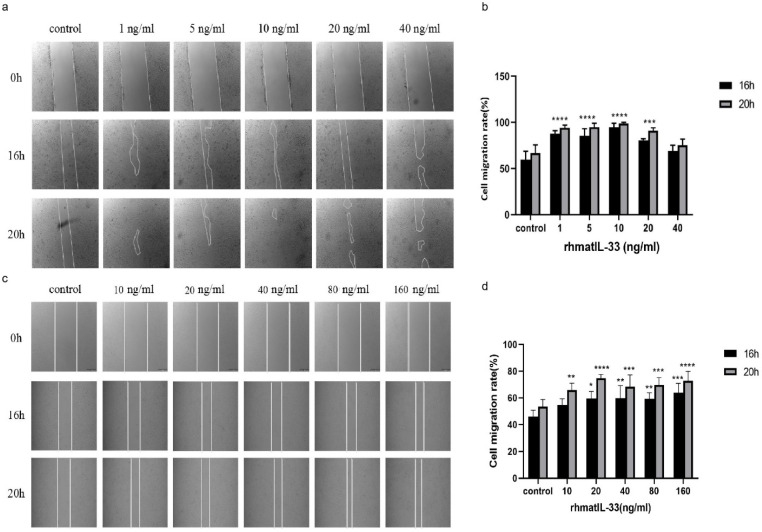
Effects of rhmatIL-33 on the migration of HSF and HaCaT cells. The wound-healing assay showed rhmatIL-33 accelerates the migration of (**a**) HSF and (**c**) HaCaT cells. (**a**,**c**) Representative images of scratched and recovered wounded areas on confluence monolayers of HSF and HaCat cells at different concentration treatments. (**b**,**d**) The quantitative evaluation and statistical analysis of cell migration rate in wound scratch assay measured by Image J software. The values are expressed as the mean ± SEM from three independent experiments. * *p*, 0.05; ** *p*, 0.01; *** *p*, 0.001; **** *p*, 0.0001.

**Figure 3 bioengineering-09-00734-f003:**
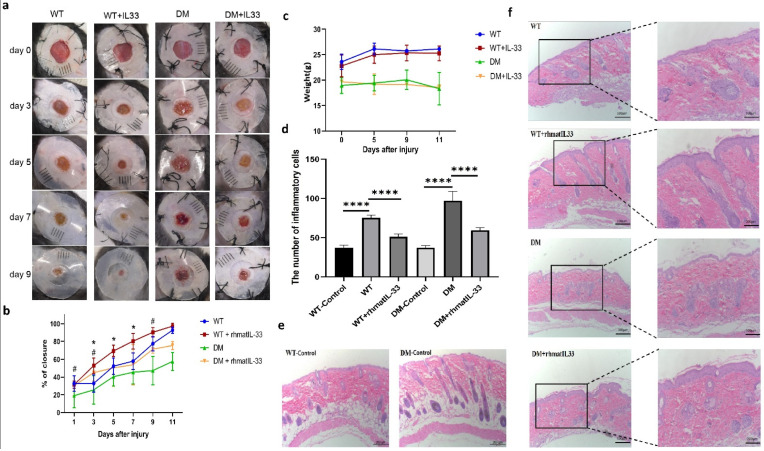
Effect of rhmatIL-33 on skin wound healing in mice. (**a**) Representative images of wounds at days 0, 3, 5, 7, and 9 post-administration. (**b**) The rate of wound closure in wounds receiving a different dose of rhmatIL-33 at the indicated times; # *p*, 0.05, for DM vs. DM + rhmatIL-33 group; * *p*, 0.05, for WT vs. WT + rhmatIL-33 group. (**c**) The body weight of mice receiving administration at the indicated times. (**d**) The number of inflammatory cells in the skin of mice; **** *p*, 0.0001. (**e**) HE staining of intact mouse skin. (**f**) H&E staining of wound sections treated with rhmatIL-33, PBS at day 5 post-administration (100×, 200×).

**Figure 4 bioengineering-09-00734-f004:**
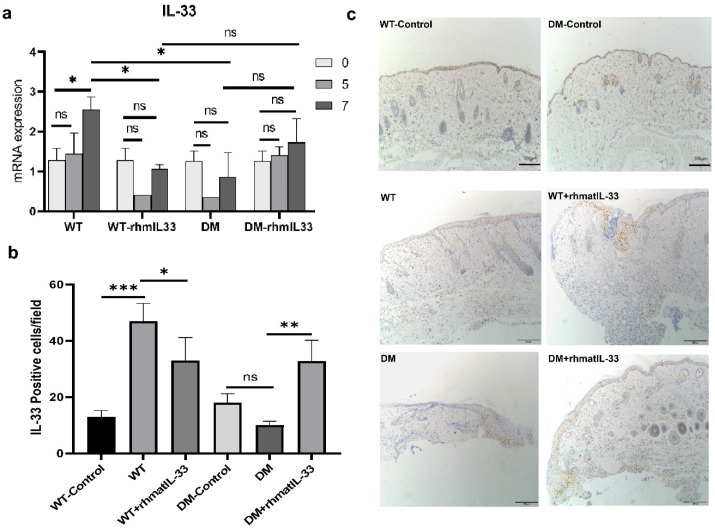
Expression of endogenous IL-33 in skin wounds. (**a**) Expression of the IL-33 mRNA in wound tissue at day 5 post-administration. (**b**) The number of IL-33 positive cells is presented in the form of a graph. * *p*, 0.05, ** *p*, 0.01, *** *p*, 0.001 and ns, not significant. (**c**) IL-33 immunohistochemical staining in the skin of mice.

**Figure 5 bioengineering-09-00734-f005:**
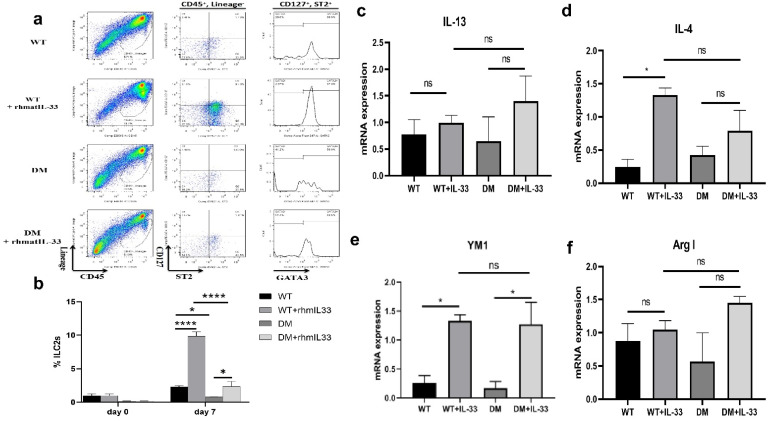
Effects of rhmatIL-33 administration on ILC2 cells in wound tissue. (**a**) Flow cytometric gating strategies for isolation and sorting of ILC2s. (**b**) Bar graphs showing the percentage of ILC2s in the mouse skin at different time points. (**c**–**f**) Gene expressions of repair-related cytokines in mouse wound skin tissue treated with rhmatIL-33 or PBS by real-time RT-PCR analyses. * *p*, 0.5; **** *p*, 0.0001 and ns, not significant.

**Figure 6 bioengineering-09-00734-f006:**
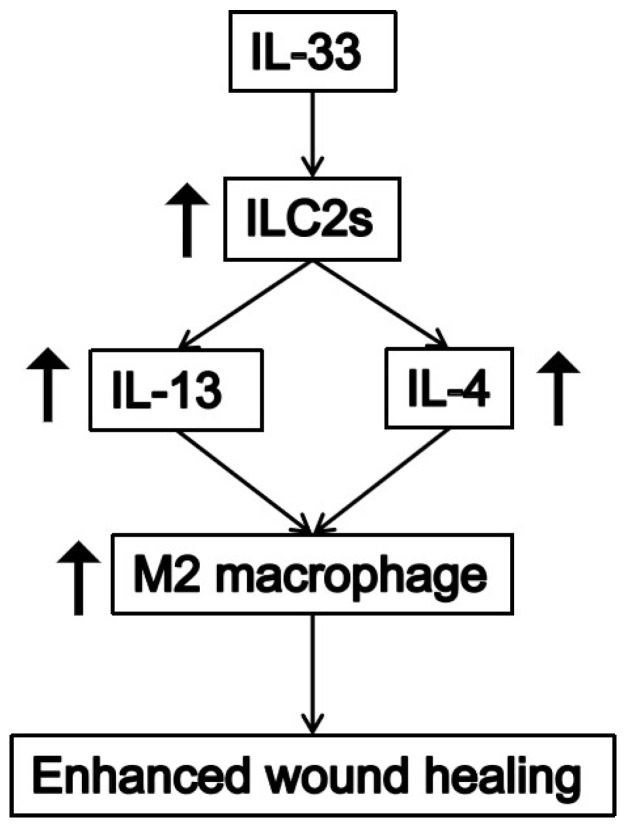
Wound repair mechanisms of IL-33 in skins. IL-33 promotes the proliferation of ILC2 cells, which leads to the up-regulation of IL-13 and IL-4. IL-13 and IL-4 further promote M2 macrophage polarization. All together, these contribute to accelerated and improved wound healing.

## Data Availability

Not applicable.

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
