# Peer review of "Recombinant Expression of Human IL-33 Protein and Its Effect on Skin Wound Healing in Diabetic Mice"

_bioengineering, 2022, doi:10.3390/bioengineering9120734_

Round 1

Reviewer 2 Report

The manuscript reports the recombinant expression of human IL-33 and its effects on diabetic wound healing. The manuscript is well presented and the conclusion is supported by the data provided.

Minor suggestions.

It will be better if the title is recombinant expression of human interleukin 33, instead of expression of recombinant human IL-33 protein. This is because the original title sounds like where the recombinant human IL-33 protein is expressed in the bacterial cells.

Space is needed between numerical value and units.

ul should be uL

it will be good to have a schematic diagram at the end to show how IL-33 improves diabetic wound healing

Figure 1A the top gel is Comassie blue staining? or anti-his? molecular marker is needed for the anti-IL33 WB image.

It is not recommended to say your proteins' purity is 100%, usually >90-95%. This is because you are using 280nm and it is actually unclear whether other wavelength can detect impurity, unless you have some MS results to show.

Any reason why higher concentration of IL33 does not promote cell migration for fibroblast cells? Any positive control being used for comparison e.g. EGF?
